

# LFC-UNet: learned lossless medical image fast compression with U-Net

Hengrui Liao and Yue Li

School of Computer, University of South China, Hengyang, Hunan, China

## ABSTRACT

In the field of medicine, the rapid advancement of medical technology has significantly increased the speed of medical image generation, compelling us to seek efficient methods for image compression. Neural networks, owing to their outstanding image estimation capabilities, have provided new avenues for lossless compression. In recent years, learning-based lossless image compression methods, combining neural network predictions with residuals, have achieved performance comparable to traditional non-learning algorithms. However, existing methods have not taken into account that residuals often concentrate excessively, hindering the neural network's ability to learn accurate residual probability estimation. To address this issue, this study employs a weighted cross-entropy method to handle the imbalance in residual categories. In terms of network architecture, we introduce skip connections from U-Net to better capture image features, thereby obtaining accurate probability estimates. Furthermore, our framework boasts excellent encoding speed, as the model is able to acquire all residuals and residual probabilities in a single inference pass. The experimental results demonstrate that the proposed method achieves state-of-the-art performance on medical datasets while also offering the fastest processing speed. As illustrated by an instance using head CT data, our approach achieves a compression efficiency of 2.30 bits per pixel, with a processing time of only 0.320 seconds per image.

# INTRODUCTION

In the field of medicine, the demand for high-quality image storage continues to rise. Medical images capture detailed information about every organ and tissue in the human body, making lossless compression techniques crucial in the medical domain. However, the images to be encoded are not simply pixel arrangements; they may contain significant noise and redundant information. Lossless compression necessitates preserving all image information completely. Unlike lossy compression, which eliminates image information to achieve compression, lossless compression aims to retain all image details. Therefore, in lossless compression, the key lies in increasing the amount of information carried by each bit, *i.e.,* reducing information entropy. In fact, according to Shannon's source coding theorem (*Shannon, 1948*), the optimal code length for a symbol is determined by $-\log_b P$, where $b$. represents the number of output codes, and $P$ represents the probability of input symbols. It is evident that designing an effective probability model for pixel values is crucial in entropy coding.

Corresponding author
Yue Li, liyue@usc.edu.cn

Early learning-based lossless compression algorithms employed deep neural networks (DNNs) as autoregressive models. These algorithms relied on the powerful capability of DNNs to estimate pixel probability distributions, with the condition of considering previous samples. For instance, methods like PixelRNN (*Van Den Oord, Kalchbrenner & Kavukcuoglu, 2016*), PixelCNN (*Van den Oord et al., 2016*), and PixelCNN++ (*Salimans et al., 2017*) performed compression pixel by pixel, predicting the probability distribution conditioned on all previous pixels. However, these approaches required neural network computations for the entire pixel count, leading to impractical inference times.

To enhance practicality, recent studies have divided encoding units into sub-blocks or entire images, rather than individual pixels (*Reed et al., 2017*; *Mentzer, Gool & Tschannen, 2020*). These methods derive probability distributions for the entire image based on lossy compression, as opposed to the previous approach of processing each pixel individually. Additionally, some research, such as L3C (*Mentzer et al., 2019*), introduced a practical compression framework utilizing a hierarchical probabilistic model. Subimages are implicitly modeled by a neural network, and each subimage is conditioned on the subimage from the previous scale. In order to further improve prediction accuracy, certain studies have taken image texture information into account. For instance, LC-FDNet (*Rhee et al., 2022*) employed an image segmentation coding strategy. They first encoded smooth regions and then encoded non-smooth regions, effectively leveraging the texture information of the image. This approach efficiently utilized prior knowledge of smooth regions when dealing with non-smooth areas, resulting in a robust predictive performance for the probability model in non-smooth regions. However, both the hierarchical inference design of L3C and the two-round inference design of LC-FDNet lead to an increase in neural network inference time.

In this study, we have developed an efficient coding framework aimed at significantly reducing encoding time while enhancing coding performance. Our approach combines the features of U-Net (*Ronneberger, Fischer & Brox, 2015*) and captures residual features more effectively through skip connections. Considering the characteristic concentration of residual distributions, we employed a weighted cross-entropy loss function during training to address the issue of non-uniform residual sample distribution. In terms of time efficiency, Our network can extract all residual probabilities and residual values for a single subimage in one inference, greatly reducing the network inference time.

Our compression framework, as illustrated in Fig. 1, primarily consists of two components: the image decomposer and the learning-based compressor (LFC-UNet). Firstly, the image decomposer divides the input image into multiple subimages. One group of subimages undergoes processing through a traditional lossless compressor. Subsequently, the learning-based compressor takes the encoded subimages as input. This compressor first utilizes a neural network for a single inference, obtaining the predicted pixels and residual probabilities. It then predicts the difference between predicted and actual pixels, generating residuals. Notably, residuals are floating-point numbers that can be both positive and negative. To ensure positive integers, we perform an addition operation of 255 on residuals, followed by rounding to the nearest integer. Subsequently, the residual values and residual probabilities are fed into an entropy coder for compression. Through

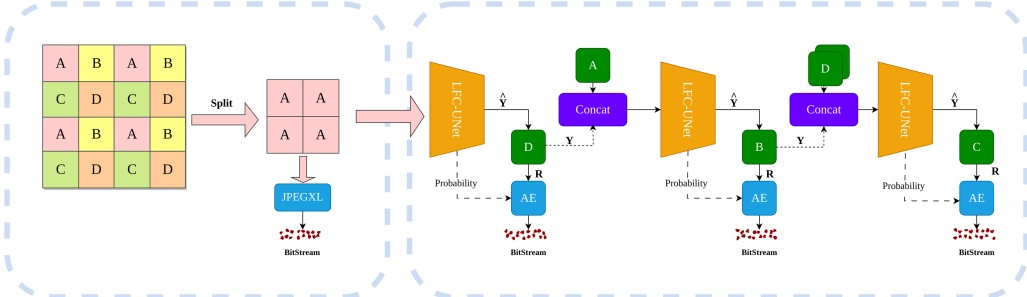

**Figure 1** **Overview of image compression process.** The image is initially processed by the left image decomposer, grouping each pixel into one of the categories: A, B, C, or D. The initial subimage 'A' undergoes compression using JPEG-XL, while the remaining subimages are compressed using the neural networks on the right side. These neural networks take previously encoded subimages as input, predicting the subimage $\hat{Y}$ and its probability estimates. The difference between $\hat{Y}$ and the actual subimage Y yields the residual R, which is then compressed using an entropy coder along with the probability estimates. Dashed lines indicate the flow of information transmission.

this design, Our method has achieved significant improvements in encoding speed and enhanced encoding performance. The main contributions are summarized as follows:

- We propose a lossless image compression framework that utilizes encoded subimages to provide spatial positioning and image feature information. Adopting a method that extracts all residual probabilities and residual values for a single subimage in one inference pass can improve encoding speed.
- Leveraging the skip connection features of U-Net, we more accurately predict the probability distribution. Simultaneously, addressing the concentrated nature of residual distributions, we employ a weighted cross-entropy loss function (*Rezaei-Dastjerdehei, Mijani & Fatemizadeh, 2020*), resolving the issue of imbalanced residual value categories.
- Our method not only achieves state-of-the-art performance on medical datasets but also excels in processing speed. This outstanding performance positions our method as highly promising in the field of medical image processing, offering extensive prospects for applications.

# RELATED WORK

## Traditional image codecs

In the field of image compression, traditional lossless image encoders are commonly categorized into transform-based methods and prediction-based methods. These two categories possess unique characteristics, offering flexible solutions for various application scenarios. In this section, we will introduce several representative traditional lossless image encoders: JPEG-XL (*Alakuijala et al., 2019*), PNG (*Boutell, 1997*), WebP (*Version, WebEngines Blazer Platform, 2000*), and BPG (*Fabrice Bellard contributors, 2023*). These encoders exhibit significant features in terms of image quality, compression efficiency, and application domains.

JPEG-XL is a transform-based lossless image encoder. Its optimization strategy includes the use of variable-sized DCT (*Khayam, 2003*) blocks, allowing better adaptation to different frequency components of the image. Local adaptive quantization further enhances image quality, while additional DC coefficient prediction modes improve prediction accuracy. The adoption of asymmetric numeral systems (ANSE) enhances coding efficiency, achieving better compression ratios.

PNG, on the other hand, employs prediction-based lossless image encoding. It represents each pixel by the difference between the current pixel and its surrounding neighboring pixels through differential prediction. This differential data is encoded using various prediction filters, achieving lossless compression. PNG maintains high image quality while offering a high compression ratio, making it suitable for scenarios requiring high-quality image transmission, such as digital art and medical imaging fields.

WebP is a lossless image compression format developed by Google, combining prediction-based methods with techniques such as context modeling. It strikes a balance between small file sizes and high-quality images and finds widespread usage in web images and mobile applications. WebP's features make it an ideal choice for web image transmission, providing users with faster loading speeds and a better user experience.

BPG is an emerging lossless image compression format that combines prediction-based methods with advanced compression techniques, such as prediction filtering and motion compensation. BPG achieves relatively small file sizes while preserving high-quality images, making it popular in scenarios where maintaining image quality while reducing file size is crucial. These encoders offer diverse and efficient choices for lossless image compression in different domains.

In summary, traditional image codecs commonly employ two main techniques for lossless compression: transformation and prediction. These techniques aim to reduce redundancy in image information and enhance coding efficiency. Transformation, exemplified by methods such as discrete cosine transform (DCT) used in applications like JPEG-XL, WebP, and BPG, alters the representation of image data in the spatial domain to the frequency domain. This organized presentation of frequency components makes image data more compact, forming the basis for subsequent prediction steps, all while ensuring reversibility to avoid information loss. Prediction, on the other hand, leverages regularities or patterns in data to forecast future data. It achieves more efficient compression by encoding residuals that eliminate redundant information. For instance, PNG utilizes predictive methods with filtering based on the idea of differential encoding, reducing redundancy by computing the difference between the current pixel value and adjacent pixel values. Additionally, JPEG-XL employs DC coefficient prediction, analyzing information from neighboring blocks or rows to predict the DC coefficients of the current block. This strategy reduces the dynamic range of DC coefficients, thereby improving the coding efficiency for low-frequency information in the image.

## Learning-based image compression

In early methods, image encoding was performed on a pixel-by-pixel basis, where the compression of each pixel relied on previously encoded pixels. For example, PixelRNN and

PixelCNN modeled pixels as conditional products, expressed as $p(x) = \prod p(x_i|x_1, \ldots, x_{i-1})$, where $x_i$ represent a single pixel. Subsequently, PixelCNN++ introduced several important optimizations on the original PixelCNN model, including improvements in generation efficiency and image quality. PixelCNN++ employed a segmental logistic mixture model for layer-wise modeling, introduced residual connections, and utilized techniques like multiscale generation to enhance both generation speed and image quality. However, it is worth noting that PixelCNN++ still involved network computations for each pixel, resulting in impractical inference times.

To enhance practicality, subsequent research proposed several improved methods. For instance, L3C introduced a practical compression framework utilizing hierarchical probability models. subimages were implicitly modeled through neural networks and aided by a feature extractor for prediction tasks. In contrast to the pixel-wise predictions of PixelRNN and PixelCNN, L3C predicted probabilities for all pixels, achieving more than two orders of magnitude acceleration.

Recent studies like LC-FDNet took into account the helpfulness of texture information in improving probability predictions. It adopted a method of predicting image splitting, dividing the image into high-frequency and low-frequency parts. It encoded the low-frequency region first, then utilized prior information from the low-frequency region to predict the high-frequency region, thereby improving the effectiveness of probability prediction.

In the field of medicine, *Wang et al. (2023)* and colleagues have proposed a new perspective. They divided the image into two types of subimages: the most significant bytes (MSB) subimage, representing the high-order bits, and the least significant bytes (LSB) subimage, representing the low-order bits. Due to the significantly lower bit rates required for compressed MSB subimages compared to compressed LSB subimages, they chose conventional methods to encode MSB. Simultaneously, they used MSB as input for a neural network to predict LSB. Subsequently, they encoded LSB.

In learning-based image compression methods, the primary research goal is to find an approach that achieves a balance between probability prediction accuracy and prediction time. From the initial autoregressive methods, predicting pixel by pixel, to the layered context (L3C) employing implicit modeling for hierarchical prediction, and further to segmented prediction methods proposed by researchers such as LC-FDNet and Kai Wang, it is evident that researchers have consistently pursued this balance.

## METHOD

The overall process of our method is illustrated in Fig. 1, given an input image $x \in R^{W \times H \times 1}$, where $W$ represents the width of the image, $H$ represents the height of the image, and 1 represents the number of channels in the grayscale image. We utilize an image decomposer to break down the image into subimages. Specifically, we divide the input image into four subimages $x_s \in R^{\frac{W}{2} \times \frac{H}{2} \times 1}$, where $s$ represents spatial position indices($s \in \{a, b, c, d\}$). First, the subimage $x_a$ undergoes compression using a conventional compression algorithm.Then, the encoded subimage $x_a$ serves as input to a neural network, producing predictions for

pixels and residual probabilities for the subimage $x_d$. The residuals are obtained by taking the difference between the predicted pixels from subimage $x_d$ and the actual pixels of subimage $x_d$ which are then entropy coded. Similarly, the process for subimages $x_d$ and $x_c$ is identical to that of subimage $x_d$.

## Image decomposer

In this stage, we divide the image into subimages following the pattern illustrated in Fig. 1, where pixels in odd rows and odd columns are categorized as subimage A, pixels in odd rows and even columns are categorized as subimage B, and so on, resulting in four subimages. Subsequently, we perform lossless compression on subimage A pixels. We chose to use JPEG-XL to compress the subimage A images. This choice was made after a comprehensive comparison between traditional lossless codecs and learning-based ones, where we found that JPEG-XL yielded the best results. In the realm of traditional lossless codecs, JPEG-XL demonstrated superior encoding efficiency. Regarding learning-based lossless codecs, except for Iwave (*Ma et al., 2020*), other methods either performed worse than JPEG-XL or did not show a significant advantage over it. However, the runtime of Iwave was deemed unacceptable. For example, on an image with dimensions $768 \times 768$, in an environment with GeForce GTX 3060 and Intel Core i7- 12700, JPEG-XL took approximately 1.1 s, while Iwave's processing time extended to a lengthy 160.4 s.

## Learning-based compressor

In this section, we introduce the architecture of LFC-UNet as depicted in Fig. 2, which provides a detailed overview of the details outlined in Fig. 1. The objective of LFC-UNet is to compress the $N$-th subimage Y given the input $x_{in}$, which includes the $N$-1 previously compressed subimages Y. As described in Fig. 1, we employ three independent LFC-UNet networks, each tailored to handle a different number of input subimages. Consequently, these networks do not share parameters, as each of them focuses on a specific subimage.

**Preprocessing**: During the preprocessing stage, we first employ a neural network to extract features from the input subimages $x_{in} \in R^{\frac{W}{2} \times \frac{H}{2} \times N}$, where $N$ represents the number of previously compressed subimages ($N \in \{1, 2, 3\}$). Subsequently, the extracted features are transformed into an intermediate matrix in the latent space, referred to as the "Middle Matrix." This matrix serves as the input for subsequent probability predictions and image predictions.

**Subimage prediction**: In Fig. 2, "Predict Image" represents the prediction of the "Real Image." After obtaining this prediction, we calculate the residual image residual = Real Image - Predict Image, which signifies the difference between the real subimage and the prediction. As residuals might include negative and floating-point numbers, we add 255 to all residuals and then quantize them (round to the nearest integer), ultimately constraining the range of residuals within integers from 0 to 511. This quantized residual, denoted as Q, is then passed on to the entropy coder for further processing.

**Probability distribution prediction**: In Fig. 2, "Probability" represents the probability prediction of the quantized residual image Q. In this step, we use the U-Net architecture shown in Fig. 3 for prediction. The input Middle Matrix undergoes four down process

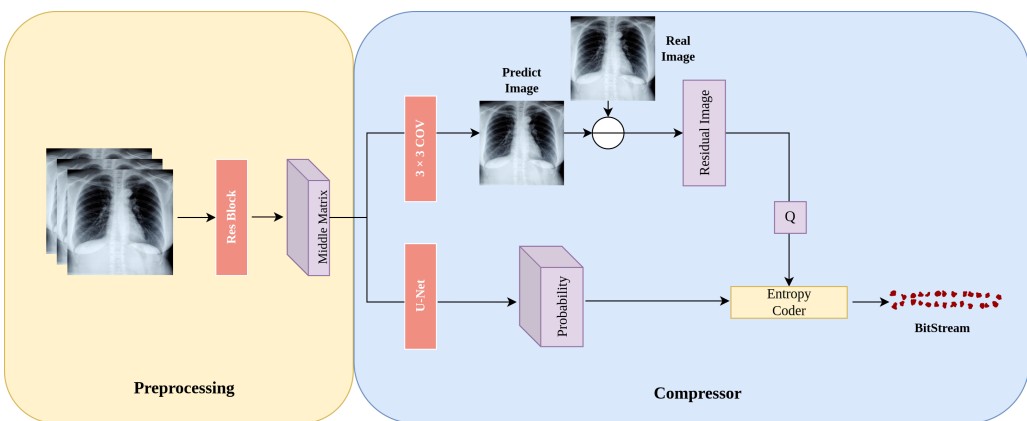

**Figure 2  Details of the LFC-UNet implementation.** The input subimages undergo preprocessing on the left to obtain a latent space represented as the Middle Matrix. The compressor on the right utilizes the Middle Matrix as input to generate probability estimates and Predict Image. The residual image is obtained by taking the difference between Predict Image and Real Image, followed by quantization of the residual. The quantized residual, along with the probability estimates, is fed into an entropy coder for encoding.

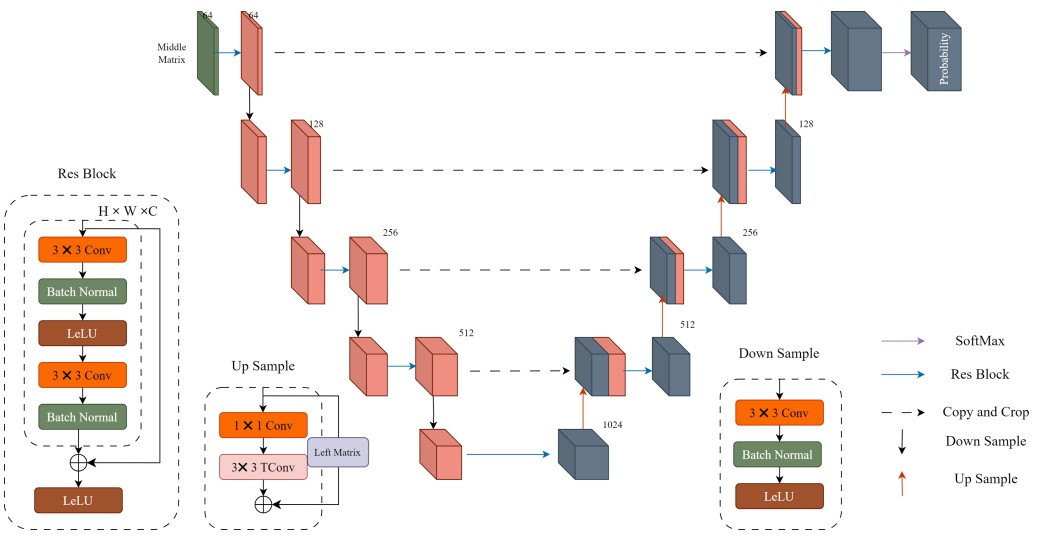

**Figure 3  Details of the U-Net implementation.** Each res block and down sample constitute a downward process, while each res block and up sample constitute an upward process. The middle matrix undergoes four downward processes and four upward processes, resulting in a 511-dimensional output. Finally, the output is transformed into probability estimates using the SoftMax function.

and four up process operations, and finally, through the Softmax function, the predicted probability distribution ensures that the sum of probabilities at each position equals 1. This results in the probability prediction of the quantized residual Q.

## Loss function

In this section, we introduce the loss functions used in the training of LFC-UNet. It consists of two components: subimage prediction loss and probability prediction loss.

**Subimage prediction loss**: We define the subimage prediction loss as the absolute difference between the predicted pixel values and the original pixel values. Mathematically, it is defined as follows:

$$L_{SPL}(y, \hat{y}) = \frac{1}{N} \sum_{i=1}^{N} |y_i - \hat{y}_i|$$

Where $N$ represents the total number of pixels in the image, $y$ represents the original pixel values, and $\hat{y}$ represents the predicted pixel values by the neural network.

**Probability prediction loss**: In the probability loss function, we employ weighted cross-entropy, defined mathematically as follows:

$$L_{PPL}(p, \hat{p}) = -\frac{1}{N} \sum_{i=1}^{N} \sum_{j=1}^{C} w_j \cdot p_{i,j} \cdot \log(\hat{p}_{i,j})$$

where $N$ represents the total number of pixels in the image, $C$ represents the number of classes, denoting the magnitude of residual values in the article, with a range from 0 to 510, $p_{i,j}$ represents the probability that the $i$-th pixel actually belongs to class $j$. More precisely, $p_{i,j}$ is equal to 1 if the $i$-th pixel belongs to class $j$, and 0 otherwise. $\hat{p}_{i,j}$ is the predicted probability of the $i$-th pixel belonging to class $j$ by the model, and $w_j$ is the weight for class $j$. This is a variant of cross-entropy loss, commonly used to address class imbalance issues. In our experiments, the neural network predicts pixel values, where 90% of the predictions differ from the actual pixel values by no more than 50. This results in a large concentration of label values around 255. Therefore, conventional cross-entropy loss performs sub-optimally in such situations. Altogether, we train our network with the loss:

$$L = \lambda_{SPL} \cdot L_{SPL} + \lambda_{PPL} \cdot L_{PPL}$$

where $\lambda_{SPL}$ and $\lambda_{PPL}$ are the balancing hyperparmeters. In our experiments, we set both $\lambda_{SPL}$ and $\lambda_{PPL}$ as 1.We opt for the normalization of the hyperparameters of the loss functions for each module to avoid unnecessary model optimization, thereby facilitating a more straightforward training and evaluation of the model. This approach represents a balance between parameter tuning and the pursuit of optimal performance.

# EXPERIMENT AND ANALYSIS

## Experimental setup
### Dataset

Our research has achieved significant advancements in the field of medical image compression. Recognizing the absence of standardized test datasets tailored specifically for medical image compression, we took the initiative to curate a dataset, allowing us to comprehensively evaluate our model's performance. This custom dataset comprises images sourced from four distinct medical datasets: the COVID RADIOGRAPHY Database

(*Rahman et al., 2021*; *Chowdhury et al., 2020*), COVID-19 Lung CT (*Zhao et al., 2020*), Head CT (*Kitamura, 2018*) and Lumbar Spine CT (*Heywhale, 2023*). To ensure diversity, we randomly sampled 50 chest X-ray images from the COVID RADIOGRAPHY Database, 50 pulmonary CT images from COVID-19 Lung CT, 50 brain CT images from the Head CT dataset and 50 Lumbar Spin CT images from the Lumbar Spin CT dataset.

In addition to our medical image testing, we conducted comprehensive evaluations in the domain of natural images. For this purpose, we selected 100 images from the COCO (*Lin et al., 2014*) natural image dataset. Notably, these COCO images were originally in RGB format, each pixel composed of red (R), green (G), and blue (B) channels. To simulate real-world scenarios, we transformed these color images into single-channel grayscale images before subjecting them to testing. This step added complexity to our model's task, as it had to maintain image quality despite reduced information.

By including both medical and natural images, including grayscale natural images, in our evaluation, we not only verified our model's performance in medical contexts but also showcased its robustness and adaptability across diverse image types.

## Training

To ensure both data diversity and model generalization, we opted to train our network utilizing the Flickr2k (*Lim et al., 2017*) dataset, which consists of 2650 high-quality images. In our training process, we took steps to enhance the variability of the dataset. Specifically, we randomly extracted image patches measuring $128 \times 128$ pixels from the original color images. Importantly, to align with our research focus on grayscale images, we converted the RGB images from the Flickr2k dataset into grayscale before training the network.

Throughout the training, we employed the Adam (*Kingma & Ba, 2014*) optimizer with a batch size of 24. The training regimen spanned 2,000 epochs, ensuring that the network had the opportunity to learn intricate patterns within the data. To strike an optimal balance between exploration and exploitation, we set the initial learning rate at $1 \times 10^{-3}$. To prevent overfitting and encourage the network to generalize well, we implemented a learning rate decay strategy, halving the rate every 500 epochs. These meticulous training procedures were instrumental in refining our model's ability to handle grayscale images effectively, contributing to its robustness and adaptability in diverse contexts.

It is noteworthy that the configuration of hyperparameters is not fixed and is adjusted based on empirical observations and the characteristics of the actual hardware devices. Regarding the setting of the learning rate, we referred to recommendations from other researchers (*Mentzer et al., 2019*; *Rhee et al., 2022*; *Tissier et al., 2023*; *Wang et al., 2023*) regarding network configurations. As for the batch size and the number of training epochs, careful consideration was given to the size of the GeForce GTX 3060 VRAM and the scale of the training dataset.

## Evaluation

In our study, we conducted a comprehensive comparison of our method against both learning-based and non-learning-based compression algorithms. Specifically, we compared our approach with several traditional lossless image codecs, including PNG, JPEG2000 (*Christopoulos, Skodras & Ebrahimi, 2000*), WebP, BPG, FLIF (*Sneyers & Wuille, 2016*), and

**Table 1  Summary of compression performance comparison.** Our method compared with other non-learning and learning-based codecs on the dataset. Performance is measured in BPP. The best performance is highlighted in bold, the second best performance is indicated with an asterisk (*).

| Method | COVID-19 RADIOGRAPHY | Lung CT | Head CT | Lumbar Spine CT | COCO |
|---|---|---|---|---|---|
| PNG | 3.17 + 22.7% | 4.32 + 26.9% | 3.70 + 37.8% | 3.53 + 21.2% | 4.27 + 14.8% |
| JPEG2000 | 4.60 + 46.7% | 4.52 + 30.1% | 4.66 + 50.6% | 4.16 + 33.2% | 5.10 + 28.6% |
| WebP | 2.87 + 14.6% | 3.52 + 10.2% | 3.00 + 23.3% | 3.30 + 15.8% | 4.07 + 10.6% |
| FLIF | 2.69 + 9.2% | 3.36 + 6.0% | 2.63 + 12.5% | 3.14 + 11.5% | 3.91 + 6.9% |
| JPEG-XL | 2.51 + 2.4% | 3.43 + 7.9% | 2.61 + 11.9% | 3.06 + 8.9% | 3.78 + 3.7% |
| L3C | 3.57 + 31.4% | 4.21 + 24.9% | 3.63 + 36.6% | 4.98 + 44.2% | 4.68 + 22.2% |
| LC-FDNet | 2.48* + 1.2% | 3.21* + 1.6% | 2.39* + 3.8% | 2.84* + 2.1% | 3.68* + 1.1% |
| Our | **2.45** | **3.16** | **2.30** | **2.78** | **3.64** |

JPEG-XL. Additionally, we evaluated learning-based methods such as L3C and LC-FDNet. The evaluation of compression performance was based on bits per pixel (BPP), where lower BPP values indicate superior compression. Encoding speed was assessed in seconds per picture (SPP), where lower SPP values signify better compression efficiency.

In the comparison experiments related to encoding performance, we made modifications to the network architecture to enable lossless compression of single-channel images since L3C cannot directly compress single-channel images.

During the statistical analysis, it is crucial to note that we excluded the additional time and bitrate required by LC-FDNet for encoding in the UV channel. This exclusion was justified as single-channel images only contain luminance (Y) information and lack chrominance (UV) channels.

## Compression result

Table 1 presents the comparative results obtained from the evaluation dataset as described. It is evident that our method excels in terms of performance, whether dealing with medical images or natural images in grayscale format, outperforming existing traditional codecs as well as learning-based codecs.

Notably, in datasets such as Lung CT, Head CT, and Lumbar Spine CT, there is a significant performance gap between traditional encoders and learning-based encoders. This disparity can be attributed to the broader grayscale range typically found in CT images, showcasing density variations within tissues, while natural images usually operate within the dynamic range of a camera, exhibiting a narrower grayscale spectrum. Traditional encoders often struggle with processing images featuring such extensive grayscale ranges. Our learning-based approach outperforms other algorithms, indicating its high practicality in the medical field.

Furthermore, our method achieves state-of-the-art performance in diverse grayscale natural images, demonstrating the model's generalizability and robustness.

Specifically, when compared to the best learning-based methods: In the Chest X-ray dataset, our method surpasses LC-FDNet by 1.2%. In the Lung CT dataset, our method outperforms LC-FDNet by 1.6%. In the Head CT dataset, our method exceeds LC-FDNet

**Table 2** Summary of compression time comparison on CPU. Our method was compared with other learning-based codecs on the dataset using CPU. Performance was measured in average encoding time per image (SPP). The best performance was highlighted in bold, the second best performance was indicated with an asterisk (*).

| Method | COVID-19 RADIOGRAPHY | Lung CT | Head CT | Lumbar Spine CT | COCO |
|---|---|---|---|---|---|
| L3C | 1.27 + 39.3% | 2.30 + 35.2% | 3.43 + 50.7% | 3.46 + 41.3% | 3.51 + 36.8% |
| LC-FDNet | 0.947* + 18.6% | 1.74* + 14.4% | 2.14* + 21.0% | 2.63* + 22.8% | 2.89* + 23.2% |
| Our | **0.771** | **1.49** | **1.69** | **2.03** | **2.22** |

**Table 3** Summary of compression time comparison on GPU. Our method was compared with other learning-based codecs on the dataset using GPU. Performance was measured in average encoding time per image (SPP), with the best performance highlighted in bold.

| Method | COVID-19 RADIOGRAPHY | Lung CT | Head CT | Lumbar spine CT | COCO |
|---|---|---|---|---|---|
| LC-FDNet | 0.155 + 14.8% | 0.285 + 15.4% | 0.409 + 21.8% | 0.427 + 21.8% | 0.441 + 22.2% |
| Our | **0.132** | **0.241** | **0.320** | **0.334** | **0.343** |

by 3.8%. In the Lumbar Spine CT dataset, our method surpasses LC-FDNet by 2.1%. In the COCO dataset, our method outperforms LC-FDNet by 1.1%.

Additionally, when compared to the best traditional methods: In the Chest X-ray dataset, our method outperforms JPEG-XL by 2.4%. In the Lung CT dataset, our method exceeds FLIF by 6.0%. In the Head CT dataset, our method surpasses JPEG-XL by 11.9%. In the Lumbar Spine CT dataset, our method outperforms JPEG-XL by 8.9%. In the COCO dataset, our method excels by 3.7% compared to JPEG-XL.

## Time result

Our setup consists of GeForce GTX 3060, Intel Core i7-12700, and 32GB of RAM. Table 2 presents the results of time comparison conducted on the specified evaluation dataset using CPU, with the unit measured in seconds. It is worth noting that, in the CPU comparison experiments, we exclusively compared our model with other learning-based methods and did not involve comparisons with traditional methods. This decision was made because traditional compression algorithms, such as JPEG XL, have already been optimized for CPU performance. These optimizations include the application of new instruction sets, multi-threading optimization, and adaptation to CPU hardware characteristics. In contrast, specialized instruction sets for neural network computations, such as CUDA, are designed explicitly for GPUs, and neural network designs involve a significant amount of matrix operations, which are not the strengths of CPUs. Therefore, we believe that conducting time comparisons with traditional methods lacks practical significance, and as a result, we chose not to perform such comparisons.

In Table 3, we exhibit the time comparison results using GPU on the same evaluation dataset, also measured in seconds. It is essential to note that L3C cannot operate on NVIDIA 30-series GPUs; therefore, our comparison was exclusively conducted with LC-FDNet.

In terms of the encoding method, we employed a serial encoding approach, encoding images one by one. Specifically, encoding input images of size $x \in R^{B \times 1 \times W \times H}$, where $B$ represents the batch size, $W$ represents the image width, and $H$ represents the image length. In this experiment, we set $B$ to 1. In practical applications, the batch size will be

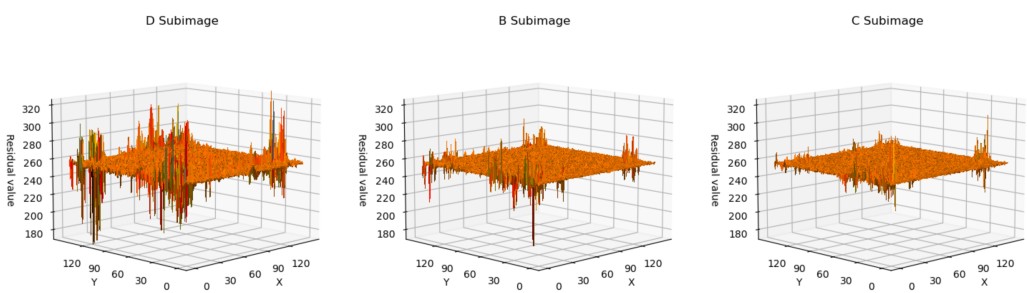

**Figure 4  The 3D plot of residual distributions.** The *X*-axis represents the length of the residual plot, the *Y*-axis represents the width of the residual plot, and the *Z*-axis represents the residual values at the position (x, y) on the residual plot.

adjusted based on the specific GPU memory configuration, potentially leading to even faster processing speeds.

Evidently from the data, our method significantly outperforms L3C and LC. This disparity can be attributed to our model's remarkable ability to obtain all probability estimates and predicted pixels in a single inference, whereas L3C's hierarchical design or LC's high-low frequency decomposition necessitates multiple inferences.

## Residual analyze

In Fig. 4, we utilized a 3D surface plot to illustrate the data, where the X and Y axes represent the two-dimensional coordinates of the residual plots, and the Z axis represents the Residual values. This graphical representation intuitively displays the residual distributions of the three subimages. Taking LungCT as an example, it can be observed that the residual values of all subimages are concentrated around 255. This indicates that the neural network has successfully transformed widely distributed pixel values into concentrated residual values. Additionally, it is noteworthy that the residuals of subimage D exhibit significant fluctuations, whereas the residual distributions of subimage C are relatively smooth. This implies that as the encoded subimages provide more information, the network's accuracy in pixel estimation gradually improves. This observation effectively explains the sequence of improved compression efficiency as $d \rightarrow b \rightarrow c$. For encoded content that is relatively smooth, the neural network can more easily predict its probability estimation. Conversely, for content with significant fluctuations, the network struggles to obtain accurate probability estimates.

In Fig. 5, we presented the data using a histogram. Here, the *X*-axis represents the residual values, and the *Y*-axis represents the frequency of occurrence corresponding to each residual value. This histogram precisely illustrates the distribution of residuals and the frequency of different residual values. This further demonstrates that with an increase in the encoded content, the network's estimation of pixel values becomes more accurate.

## Residual probability analyze

We replaced the original images and selected relatively smooth residuals for encoding. Although this aids in the network's probability estimation, it also leads to an excessive

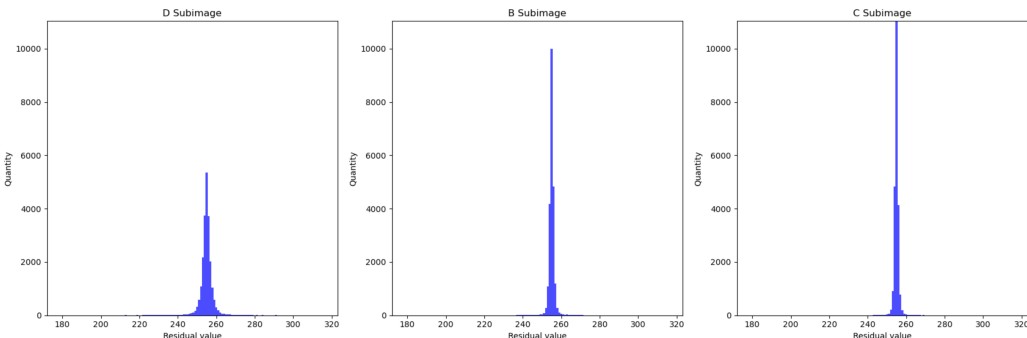

**Figure 5** **Histogram of residual distributions.** The $X$-axis represents the residual values, and the $Y$-axis represents the frequency of occurrence corresponding to each residual value.

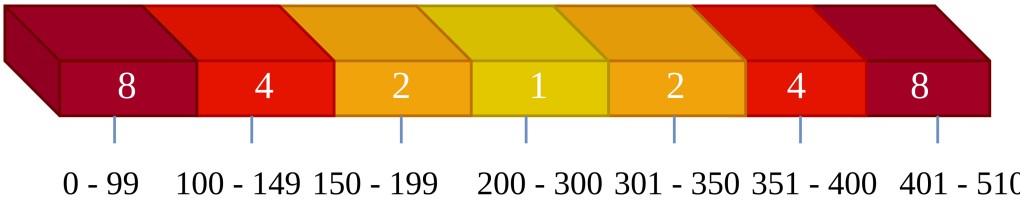

**Figure 6** **Weighted settings diagram for the cross-entropy loss function.** The numbers inside the squares represent the weights, and the numbers below the squares indicate the corresponding ranges for these weights.

concentration of residual types, as demonstrated effectively in Fig. 6. Consequently, using a conventional cross-entropy loss function would make it challenging for the neural network to learn probability estimates for residual types that occur less frequently. To address this issue, we adopted a weighted cross-entropy loss function. In this experiment, we set the weights as illustrated in Fig. 6. Within the range of 0–99, representing cases where the absolute difference between the predicted and actual values falls between 156–255, these samples are extremely rare, so we set their weight to 8. In the range of 200–300, indicating cases where the absolute difference between predicted and actual values falls between 0–55, these samples are very common, so we set their weight to 1. The weights for other intervals are set in a similar manner, following this logic.

## Loss function ablation study

We conducted ablation experiments on the Head CT dataset. In this experiment, Model 1 utilized a conventional cross-entropy loss function, while Model 2 employed a weighted cross-entropy loss function. Simultaneously, the neural network was trained in the same manner on the Flickr2k dataset. We evaluated the compression performance of subimages using these two models, measured in BPP, where smaller BPP values indicate more accurate probability estimation by the network. For detailed experimental results, please refer to Table 4. It is evident that the model using the weighted cross-entropy loss function outperformed the model using the conventional cross-entropy loss function by 3.5%.

**Table 4** **Overview of models compression performance comparison.** Our models were individually encoded on subimages B, C, and D. Performance is measured in BPP.

| Subimage | Model1 | Model2 |
| --- | --- | --- |
| B | 0.41 | 0.39 |
| C | 0.37 | 0.35 |
| D | 0.65 | 0.64 |
| Total | 1.43 | 1.38 |

## CONCLUSIONS

This study makes two significant contributions to the field of lossless compression. Firstly, it introduces an innovative approach using a weighted loss function to address the issue of overly concentrated residuals. This effectively handles the uneven distribution of residuals, thereby enhancing the performance of the model. Secondly, the successful extension of the U-Net network module as a part of the residual probability estimation into the domain of lossless compression demonstrates the multifunctionality of U-Net across different tasks. Future research could consider introducing a multi-type tree image partitioning strategy, similar to VVC (*Bross et al., 2021*), to more fully leverage sub-image information, enhance the accuracy of probability estimation, and further improve the performance of the lossless compression model.

The experimental results indicate that our proposed compression method not only outperforms traditional lossless image compression methods and other learning-based methods but also excels in medical image compression, reaching state-of-the-art levels. Moreover, in terms of processing speed, our network gains a significant advantage because it can obtain all residuals and residual probability estimates for a single subimage in one inference, leading to remarkable speed improvements.

### Funding
The authors received no funding for this work.

### Competing Interests
The authors declare there are no competing interests.

### Author Contributions
- Hengrui Liao conceived and designed the experiments, performed the experiments, analyzed the data, performed the computation work, prepared figures and/or tables, authored or reviewed drafts of the article, and approved the final draft.
- Yue Li conceived and designed the experiments, authored or reviewed drafts of the article, and approved the final draft.

### Data Availability
The source code is available at Zenodo: Hengrui, L. (2023). LFC-UNet source code release. Zenodo. https://doi.org/10.5281/zenodo.10153070.

The network model and the testing dataset are available at Zenodo: Hengrui, L. (2023). Dataset and Model in the LFC-UNet Paper. Zenodo. https://doi.org/10.5281/zenodo.10052489.

This dataset contains a subset of the original dataset. It is provided for the convenience of reproducing the results presented in the article.

The original datasets are available at:

- COVID Radiography: https://www.kaggle.com/datasets/tawsifurrahman/covid19-radiography-database.

- COVID-19 Lung CT: https://www.kaggle.com/datasets/luisblanche/covidct.

- Head CT Hemorrhage: https://www.kaggle.com/datasets/felipekitamura/head-ct-hemorrhage.

- Lumbar Spine CT: https://www.heywhale.com/mw/dataset/653b2ae854f79798ff5c484b.

- COCO 2017 Val images: https://cocodataset.org/#download.

- FLICKR2K: https://www.kaggle.com/datasets/daehoyang/flickr2k.

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
