# Peer review of "LFC-UNet: learned lossless medical image fast compression with U-Net"

_PeerJ Computer Science, doi:10.7717/peerj-cs.1924_

## Round 0.1 · original submission · Minor Revisions

The paper tackles an important compression approach. Please carefully consider the reviewer comments.

**Language Note:** The review process has identified that the English language must be improved. PeerJ can provide language editing services - please contact us at copyediting@peerj.com for pricing (be sure to provide your manuscript number and title). Alternatively, you should make your own arrangements to improve the language quality and provide details in your response letter. – PeerJ Staff

Reviewer 1 ·

Basic reporting

The article is technically written correctly without ambiguity.
The literature is sufficient; however, using Wikipedia as a source of information is not recommended. Wikipedia references cannot be taken seriously; it is advisable to cite the original source or a basic bibliography on coding data.
The structure of the article is correct.
Figure 1 not include Title solely contain the Legend and rest of Figures the Title is not in PeerJ define format. There is also an ambiguity between uppercase and lowercase letters.
In Figures is well, but the letter size not is with respect a blocks size.
Tables is now presented in PeerJ format and no presente Title only legends.
Letters used by variables need has in math text format, for example “b” and “P” in lines 31 and 32;”W” and “H” in lines 138, 139, 274, 275. “N” in 162, 168, 186. “y” 186. “C” in 188, “i”, “j” in 189, “B” in 274, 275.
The equation to calculate Loss Function is confuse, the balancing hyperparameters is not considered relevant or how its use can be determined, in addition, the notation of the subimage Prediction Loss and Probability Prediction Loss is poorly written.
Notation on Table I (not Roman number) line 246
You define repeatedly bits per pixel (BPP) in all paper and one time you use (bpp)

Experimental design

It is necessary to specify why the necessary times for all the methods being compared are not analyzed.

In the paragraph, they mention ranges and weights that differ from what is presented in Figure 6. What is the correct version?

Validity of the findings

no comment

Additional comments

"It is critical to check the font format when specifying the mathematical element within the text of the document, as the correct interpretation of variables to be used depends on it; otherwise, it may cause confusion for the reader."

Cite this review as

Reviewer 2 ·

Basic reporting

The study offers a beneficial approach to learn lossless medical image fast
compression with U-Net. However, there are some revisions required to be conducted.
1. The abstract need contain the numerical result of system performance
2. Kindly review the formatting throughout the manuscript, including text style, size, formulas, tables, figures, numbers of equations, etc.
3. Minor revisions are needed to improve the style of the article, as it contains spelling and punctuation errors.
4. The authors describe the motivation and the problem well. But, during the description of the scientific background, the literature review is just a pile of information, lacking of analysis and induction.

Experimental design

1.Are authors used the images include complex and changeable conditions in experimental study?
2. The authors should provide an explanation for their choice of hyperparameters in the proposed model, such as the number of epochs, batch size, etc.
3. The authors could discuss the implications of their study for this field and suggest potential future directions for research..

Validity of the findings

seems ok

Cite this review as

---

## Round 0.2 · accepted · Accept

Upon satisfying the minor issue in the reviewer's comments' the paper will be eligible to publish.

Reviewer 1 ·

Basic reporting

The suggested changes have been made successfully.

Experimental design

The suggested changes have been made successfully.

Validity of the findings

The suggested changes have been made successfully.

Additional comments

The suggested changes have been made successfully.

Cite this review as

Reviewer 2 ·

Basic reporting

After a detailed review, it's a very goog quality paper with exclellent experimental results and analysis, the paper can be now accepted for publication.

Experimental design

It's ok.

Validity of the findings

seems ok.

Additional comments

The authors need revised the format o f references, such as in the line 384 and add the references suggested in the reponse letter.

Cite this review as